# Understanding Changes in the Topology and Geometry of Financial Market Correlations during a Market Crash

**DOI:** 10.3390/e23091211

**Published:** 2021-09-14

**Authors:** Peter Tsung-Wen Yen, Kelin Xia, Siew Ann Cheong

**Affiliations:** 1Center for Crystal Researches, National Sun Yet-Sen University, No. 70, Lien-hai Rd., Kaohsiung 80424, Taiwan; peter.yen@mail.nsysu.edu.tw; 2Division of Mathematical Sciences, School of Physical and Mathematical Sciences, Nanyang Technological University, 21 Nanyang Link, Singapore 637371, Singapore; xiakelin@ntu.edu.sg; 3Division of Physics and Applied Physics, School of Physical and Mathematical Sciences, Nanyang Technological University, 21 Nanyang Link, Singapore 637371, Singapore

**Keywords:** econophysics, financial markets, correlation filtering, minimal spanning tree, planar maximally filtered graph, topological data analysis, SGX, TAIEX

## Abstract

In econophysics, the achievements of information filtering methods over the past 20 years, such as the minimal spanning tree (MST) by Mantegna and the planar maximally filtered graph (PMFG) by Tumminello et al., should be celebrated. Here, we show how one can systematically improve upon this paradigm along two separate directions. First, we used topological data analysis (TDA) to extend the notions of nodes and links in networks to faces, tetrahedrons, or *k*-simplices in simplicial complexes. Second, we used the Ollivier-Ricci curvature (ORC) to acquire geometric information that cannot be provided by simple information filtering. In this sense, MSTs and PMFGs are but first steps to revealing the topological backbones of financial networks. This is something that TDA can elucidate more fully, following which the ORC can help us flesh out the geometry of financial networks. We applied these two approaches to a recent stock market crash in Taiwan and found that, beyond fusions and fissions, other non-fusion/fission processes such as cavitation, annihilation, rupture, healing, and puncture might also be important. We also successfully identified neck regions that emerged during the crash, based on their negative ORCs, and performed a case study on one such neck region.

## 1. Introduction

At the turn of the 20th century, Bachelier suggested in his PhD thesis that stock prices follow geometric Brownian motions and worked out some of the consequences [1]. This was a major breakthrough at that time, when few expected any theoretical understanding of the stock market. In his thesis, Bachelier assumed that the prices of term contracts follow a normal distribution. Osborn then proposed that it is the rate of return that follows a normal distribution [2]. Later, Mandelbrot and Fama independently found early evidence to suggest that this is not true, and the return distribution has fat tails better fitted by a Levy stable distribution with b=1.7 [3,4]. Mandelbrot then proposed modeling financial returns using fractional Brownian motion [5] and, later, multifractals [6]. Parallel efforts to understand the complexity of financial markets using agent-based models and evolutionary computing were also undertaken at the Santa Fe Institute by Palmer et al. [7]. Up until this point in time, physicists studied economics problems sporadically, and this body of knowledge was not yet known as econophysics.

Widely recognized to be the start of econophysics are the 1991 paper by Mantegna [8] and the 1992 paper by Takayasu and his co-workers [9]. Then, in 1995, Stanley coined the name *econophysics* during the Statphys-Kolkata conference at Kolkata, India [10]. This marked a watershed moment in the field. After 1995, more physicists worked on economic and financial problems, publishing their results and findings in physics journals. These events ushered in the field of econophysics, where physicists (and mathematicians, as well as computer scientists) brought insights from their own fields to the study of economics and finance. Over the next two decades, econophysicists witnessed several breakthroughs. The earliest success of econophysics is the application of random matrix theory (RMT, which is a statistical theory developed to explain the energy spectra of heavy nuclei) to the stock market [11,12,13,14]. In RMT, one treats noise as a kind of symmetry, and thus information represents some form of symmetry breaking. This allows physicists to discriminate between noise and signal in financial markets. The next significant milestone in econophysics was a more compelling demonstration of fat tails in return distributions by Mantegna and Stanley [15,16], and also by Mittnik et al. [17]. These two groups estimated b=1.4 for the Levy stable distribution.

Many other breakthroughs then followed, including the fitting of price time series to a log-periodic power-law (LPPL), which allowed precise predictions of market crashes [18,19], as well as the discovery of dragon kings [20] by Sornette, understanding and modeling of the Gibbs–Pareto distribution of wealth and income by Chakrabarti et al. [21] and Yakovenko [22], characterization of the actual Brownian motion in the price fluctuations [23,24], and the development of the DebtRank metric for measuring systemic risk in financial networks [25]. Other network approaches have also started appearing in econophysics recently. These include recurrence networks (RNs), visibility graphs (VGs), and transition networks (TNs). Recurrence networks were proposed by Marwan, Donner and their co-workers in 2009 [26,27] and are used to study the statistical properties of daily exchange rates [27]. Since the seminal work by Lacasa in 2008 [28], many groups have started using VGs to analyze financial time series, including exchange rates [29], stock indices across different countries [30], the macroeconomics series of China [31], and market indices in the US [32]. A recent article by Antoniades et al. [33] used the TN to investigate the Vosvrda macroeconomic model, but thus far no one has tested the approach on real financial time series data.

Other recent breakthroughs include the application of inverse statistics (IS) in finance. IS, which is deeply rooted in fluid dynamics, and related in particular to the phenomenon of turbulence, is an old yet challenging problem. For the last two decades, many concepts have been borrowed from past studies on turbulence and applied to financial problems. One of them was the use of forward statistics, which aims to answer the question “given a fixed time horizon, what are the typical returns that an investor will realize in that period?”. In addition, Jensen [34] proposed the inverse statistics, by turning the question around, to ask “for a given return on an investment, what is the typical time required to realize it?”. This latter question is no less pertinent and is more relevant to practical financial management. If IS such as the above can be computed, investors could earn market-beating profits.

Using the IS as a probe, Jensen, Simonsen, and Johansen, published a series of papers starting in the mid-2000s [35,36,37,38] to study many economic phenomena. They focused particularly on the Gain-Loss Asymmetry (GLA) in financial markets. GLA refers to the observation that, in a financial market, positive prices have different dynamics from the negative ones. After testing stock indices in the US such as the DJIA [35], Nasdaq, and the S&P 500 [37,39], those in other countries such as Austria [40], Korea [41], and 40 other world indices [39], and other instruments such as FOREX [38], mutual funds [42], it was found empirically that negative returns took shorter average times to realize compared to positive returns of the same magnitude. To explain how GLA occurs in real markets, models with a fear factor have been developed [43,44,45,46]. However, factors other than fear of loss might also explain the GLA [47]. A comprehensive survey on IS can be found in the review article by Ahlgren et al. [48].

In this Special Issue, we celebrate the breakthrough that is one of Mantegna’s crowning achievements, which is the application of the *minimal spanning tree* (MST) to unravel hierarchical structures in financial markets [49]. We will start by reviewing the essence of Mantegna’s insight, and the body of works that followed him (including the systematic embedding of cross correlations onto a hierarchy of surfaces with different genera [50]). We then describe attempts to overcome the limitations of the MST by going to hypergraph approaches [51,52,53,54]. A hypergraph is a natural extension of a graph, where instead of having each edge join only two nodes, an edge can join any number of nodes. Unfortunately, the hypergraph approach is difficult to implement starting from pairwise correlations, so we argue that the more promising approach to extract deeper insights into the hierarchical structure in financial markets is through *topological data analysis* (TDA) [55,56,57,58]. In TDA, the idea is to go beyond the concepts of nodes (0-simplex), links (1-simplex), and the network that they form to a *simplicial complex*, which can contain (k>1)-simplices as constituents.

In a recent paper [59], we demonstrated how TDA can be used to understand the topological changes that accompany market crashes. For such extreme events in financial markets, one of the key questions not well answered through the use of MSTs or planar maximally filtered graphs (PMFGs) is how the hierarchy of cross correlations between stocks re-organizes itself. In particular, an important class of topological changes is the merging between disjoint clusters (or their time reversal—the splitting of a cluster into disjoint clusters). We found, by tracking how the Betti numbers $\beta_{0}$, $\beta_{1}$, and $\beta_{2}$ change over market crashes, that $\beta_{0}$ (the number of connected components) is small at the beginning of a market crash and increases as the market crash progresses. This tells us that we have a giant connected component in the market just before the crash, and as the market crashed, this broke up into many smaller components. The nature of this breaking up can be understood in greater detail through $\beta_{1}$ (the number of “holes” in the connected components), and $\beta_{2}$ (the number of “voids” in the connected components) (see Figure 1). Based on $\beta_{1}$ and $\beta_{2}$, we realized that a particular crash occurred in two stages. In the first stage, the topology of the giant connected component became more complex, as some “voids” grew outwards to become “holes”. In the second stage, the number of “holes” decreased precipitously, presumably the result of handle-breaking events. These handle-breaking events are not simple, because the number of “voids” increases in this stage. Finally, the giant connected component broke up completely into many connected components that have simple topologies (few “holes” and “voids”).

In addition to the TDA, we found another promising approach for extending the information filtering paradigm of MSTs and PMFGs. This is through calculating discrete versions of the Ricci curvature, either the Ollivier-Ricci curvature (ORC) for networks, or the Forman-Ricci curvature (FRC) for simplicial complexes. To identify which stocks in a network or simplicial complex make up the neck or bridge region between two densely connected clusters, the naive approach would be to identify them visually. Naturally, this is laborious and inefficient. It turns out the ORC is ideal for this task, because links in the neck regions have negative ORC. More importantly, the breaking up of a manifold into two involved the stretching and narrowing of the neck region through a process called *Ricci flow*. Physical fission processes closely resemble Ricci flow, even when the objects undergoing fragmentation are networks or simplicial complexes. In such discrete Ricci flows, the ORC or FRC become more negative over time to produce finite-time singularities. Our motives in computing the ORC are threefold: First, we would like to identify the neck regions by looking for where in the network the ORCs are negative. Second, by looking at how the negative ORCs are changing, we would like to predict when we run into finite-time singularities. These are when the fissions occur. Finally, from the natures of the singularities, we would like to understand the drivers for the different fissions.

To make the case for TDA and Ricci curvature analysis, we organized our paper as follows. In Section 2, we will review applications of the MST in econophysics. In Section 3, we will explain how the PMFG can provide more details on correlations between stocks, by keeping more links than in the MST. In fact, there is a hierarchy of maximally filtered networks on closed surfaces with increasing genera (the PMFG being the simplest, on a sphere with genus g=0) that we can explore to understand the structure of correlations between stocks. Unfortunately, the algorithms for obtaining higher-order filtered networks become increasingly difficult to implement, which explains why the PMFG is not as popular as the MST. In fact, we found only one previous work that demonstrated how to filter the weighted links of an artificial complex network onto a torus (with genus g=1) [60]. In Section 4, we describe the ideas behind TDA, and suggest that this is the natural extension going beyond MST and PMFG. To make our case, we explore four toy models for fusions and fissions, and thereafter use their TDA signatures to explain non-trivial topological changes observed in the cross correlations between stocks during a market crash in the Taiwan Stock Exchange (TWSE). In Section 5, we define what Ricci curvature is for smooth surfaces, and describe how this can be generalized to discrete networks and simplicial complexes, in the form of Ollivier-Ricci curvature and Forman-Ricci curvature, respectively. We then explain why we need Ricci curvature analysis to distinguish between different stages of fission processes that are topologically equivalent, before demonstrating this power for one of the toy models. Finally, we use the Ollivier-Ricci curvature to analyze a sequence of PMFGs obtained from the cross correlations of TWSE stocks in overlapping time windows leading up to the market crash of interest, before ending with a comparative case study of two neck regions. In Section 6, we present the conclusions.

## 2. The Minimal Spanning Tree

In Figure 2, we show the matrix of Pearson cross correlations
(1)$$C_{ij}=\frac{\frac{1}{T}\sum_{t=1}^{T}\left(x_{i,t}-{\bar{x}}_{i}\right)\left(x_{j,t}-{\bar{x}}_{j}\right)}{\sqrt{\frac{1}{T-1}\sum_{t^{\prime }=1}^{T}{\left(x_{i,t^{\prime }}-{\bar{x}}_{i}\right)}^{2}}\sqrt{\frac{1}{T-1}\sum_{t^{\prime \prime }=1}^{T}{\left(x_{j,t^{\prime \prime }}-{\bar{x}}_{j}\right)}^{2}}}$$
between 561 stocks in the Singapore Exchange (SGX) within the period January 2008 to December 2009. In Equation (Equation 1), the time series $x_{i}=\left(x_{i,1},\cdots ,x_{i,t},\cdots ,x_{i,T}\right)$ and $x_{j}=\left(x_{j,1},\cdots ,x_{j,t},\cdots ,x_{j,T}\right)$ with average ${\bar{x}}_{j}=\frac{1}{T}\sum_{t=1}^{T}x_{j,t}$ can be the daily prices, daily price differences (also known as the daily returns), or daily log-returns (which are practically identical to the daily fractional returns) of stocks *i* and *j*. Their time averages are ${\bar{x}}_{i}=\frac{1}{T}\sum_{t=1}^{T}x_{i,t}$ and ${\bar{x}}_{j}=\frac{1}{T}\sum_{t=1}^{T}x_{j,t}$. In Section 4.3, we used the daily returns for our topological data analysis. This is acceptable for short time periods, e.g., six months, because the price levels do not change by much. For longer time periods, for example, two years, as in the example associated with Figure 2, we used the daily fractional returns, so that we do not have the problem of increasing weights when the price levels become significantly higher at the end of the time period.

Before the rows and columns are reordered, it is impossible to discern any correlational structures in the SGX stocks. After reordering the rows and columns, we find the strong correlations organized into diagonal blocks, with weaker correlations between them. We also see that within the largest diagonal block in Figure 2b, the correlations are not uniform, but are further organized into diagonal sub-blocks. In hindsight, doing the reordering of rows and columns to reveal these correlational structures in the SGX was a straightforward task, since they have been shown to exist in other markets [61,62,63,64,65]. Mantegna was the first to suspect such hierarchical organizations exist in stock markets and proposed methods to elucidate such structures. Like us, Mantegna employed hierarchical clustering methods to carry out the reordering of rows and columns. However, clustering methods are based on pairwise distances, so the first problem that he had to solve was mapping the conventional Pearson cross correlations, which do not satisfy the three axioms of a distance metric, to pairwise distances. After discussions with Sornette (see Ref. 14 in [49]), Mantegna adopted the mapping
(2)$$D_{ij}=\sqrt{2(1-C_{ij})}$$
going from a cross correlation $C_{ij}$ between stock *i* and stock *j* to a pairwise distance $D_{ij}$, which satisfies the *strong triangle inequality*$D_{ij}\leq \max\left\{D_{ik},D_{kj}\right\}$. Mantegna then investigated the correlational structures in the component stocks of the Dow Jones Industrial Average (DJIA) and Standards and Poors 500 (S&P 500) indices, using single-linkage hierarchical clustering. Based on these results, Mantegna argued that US stocks do not react equally strongly to the various economic factors, but do so in groups synonymous with those discovered by random matrix theory [66]. This corroboration between Mantegna’s 1999 MST paper and Plerou et al.’s 1999 RMT paper was an important discovery at that time.

However, the greatest impact of this 1999 paper was the use of the minimal spanning tree (MST) as a caricature of the correlational structures between stocks. A *tree* is a graph with no cycles, and the MST was introduced as early as the 1950s as a special subgraph of a weighted graph containing cycles. In Figure 3a, we show the algorithm attributed to Kruskal [68] for constructing an MST, as well as an example in Appendix B. Following Mantegna’s lead, many others (including ourselves) started publishing papers on the MSTs of different markets in, for example, the US [69,70,71,72,73,74,75], UK [76], Korea [77,78], Japan [79], China [80], India [81], Indonesia [82], and Africa [83]. We also find the MST applied to different classes of financial instruments: market indices [81,84,85,86], bonds and interest rates [87,88,89], currencies [90,91,92,93,94,95], commodities [96,97,98,99,100,101], overnight loans in an interbank network [102], housing market indices of different countries [103], to name just a few. Beyond Mantegna’s test of the temporal stability of the MST representation (where he changed the time period slightly, recomputed the cross correlations, and drew the MST again) [69], Onnela et al. also used the MST to visualize the progression of a market crash [70,104]. Other applications include Sun et al. [105,106] and Jiang et al. [107] using the MST to detect insider trading in stock markets, as well as Onnela et al. [70,73], Tola et al. [108], and Coelho et al. [109] using the MST for portfolio selection. The popularity of the MST in econophysics should be clear from this quick survey, and interested readers can refer to the reviews [110,111] for even more references.

## 3. The Planar Maximally Filtered Graph (PMFG)

The successes of the MST in econophysics inspired many other network studies. For example, to understand the same finance and economics problems, many groups experimented with other types of networks [112,113,114,115,116,117,118,119]. Others, such as Chen et al. [120], experimented with artificial markets on small world networks, scale-free networks, and multilayer networks, to find noticeable differences in market sentiments on these different networks. We even found work focusing on developing complex network metrics that can be used to track the evolution of financial markets across different states (for further information, see the review by Kennett and Havlin [121]). Working more or less separately from network scientists, economists approach the network structure of financial markets from the broader perspective of *market microstructure*. The National Bureau of Economic Research has a market microstructure research group that, it says, “⋯ is devoted to theoretical, empirical, and experimental research on the economics of securities markets, including the role of information in the price discovery process, the definition, measurement, control, and determinants of liquidity and transactions costs, and their implications for the efficiency, welfare, and regulation of alternative trading mechanisms and market structures” [122]. According to a quant school [123], market microstructure deals with issues of market structure and design, price formation and price discovery, transaction and timing cost, volatility, information and disclosure, and market maker and investor behavior. In short, market microstructure is a sub-field of economics that assumes a network structure as a given in financial markets, but introduces additional economic metrics that would help policy makers regulate market dynamics. To this end, we see more network metrics used in economics, and they have become more widely accepted by traditional economists. For example, in a recent paper, Tellez et al. distinguished between secured and unsecured interbank loans, and concluded that the Katz centrality and DebtRank are appropriate measures of systemic risk for the unsecured interbank network, while PageRank is more correlated with the interest rate spread in a secured interbank network [124].

Against this backdrop, one of the most important developments following the popularization of the MST was by Tumminello et al., who took the correlation filtering approach one step further. In econophysics, the MST is typically constructed starting from the cross-correlation matrix, which has N(N−1)/2 independent components. However, the MST only keeps N−1≪N(N−1)/2 of these. These (N−1) MST links are clearly important, but we may also wonder whether some of the discarded links might be just as important. Tumminello et al. realized that we can obtain a hierarchy of filtered graphs by projecting the strongest cross correlations onto surfaces with different genera *g* [50]. The simplest such projection onto a sphere (g=0) is the *planar maximally filtered graph* (PMFG). This keeps 3N−6 links, which is more than in the MST but still small. In fact, all the MST links are contained in the PMFG. One advantage of using the MST (which is also true for the PMFG) is that we keep exactly the same number of links for the same number of nodes. This can be less biased than using a correlation threshold value because a small change in the threshold value may lead to a large change in the number of links kept. After the PMFG was introduced, we found the following econophysics papers applying it [86,125,126,127,128,129]. Unfortunately, the PMFG algorithm (see Figure 3b for said algorithm, and an example in Appendix B) is difficult to parallelize. Therefore, for larger data sets, Massara et al. developed a related algorithm called the *triangulated maximally filtered graph* (TMFG) [130].

## 4. Topological Data Analysis

In this section, we explain how to go beyond MSTs and PMFGs in our understanding of complex dynamics in financial markets by making use of methods developed for topological data analysis (TDA). In Section 4.1, we explain what the shortcomings of MSTs and PMFGs are, what we can understand and what we cannot, and why it is natural to turn to TDA. Following this, in Section 4.2 we briefly explain the ideas behind different TDA methods. We also describe three contrasting toy models for two manifolds to merge together, and a fourth toy model that is like a combination of the first three in Appendix C, before using the TDA signatures for each toy model to understand a real-world market crash in Section 4.3.

### 4.1. Why Topological Data Analysis?

In most of the MST and PMFG papers, econophysicists merely correlated the topologies of the networks obtained with events in the market, with little or no further explanation. When the MSTs or PMFGs of two successive time periods were compared, analysis is in terms of links created or deleted, but the market may have different numbers of connected components in the two time periods. A more thorough analysis would be to superimpose these connected components and the filtered graphs, to better understand the underlying reasons for link-level changes to the networks. However, when we project market cross correlations onto a MST or a PMFG, we always worry that we may be throwing out important information. Furthermore, by focusing on link-level changes, we are also implicitly assuming that changes to cross correlations can be understood in terms of pairwise interactions between stocks. Already, there are suggestions on the existence of important complex system dynamics that have to be described in terms of many-body interactions. For example, in gene expression networks, there are signs that important functions involve interactions between three or more genes [131,132,133,134]. The same is possibly also true for financial markets, but to identify such interactions, we must go beyond network descriptions of such systems.

The explanation for complex topological changes to the cross correlations between stocks lies ultimately with overlapping portfolios [135,136,137,138,139,140]. Simply put, each entity on the market owns multiple stocks, and because there are more entities than there are stocks, their portfolios necessarily overlap. Even for this bipartite system of entities and stocks, a network description would be an over-simplification. Based on the signals it receives and is capable of processing, an entity periodically optimizes its portfolio by buying and selling stocks. These trading activities generate signals for other entities in the market, who then react to optimize their own portfolios. These interactions at the portfolio level are not open information, but we can observe changes to the prices of stocks, and hence the cross correlations that these interactions produce. Over time, portfolios may accumulate so many changes that the cross correlations between stocks at different times become topologically distinct. Signatures of these topological changes can be seen in the MST [73,104,141] and PMFG [75,86,142] representations.

In their seminal work, Tumminello et al. explained how to project more and more cross correlations onto the surfaces of manifolds with increasing genera [50]. By keeping more links, we keep more of the information in the cross correlations. At the same time, we admit more complex groups (such as simplices) of cross correlations. Then, instead of asking about degree distributions and hubs, we can examine the distribution of *k*-simplices in the network, and how different simplices are connected to each other. The network obtained from the projection of cross correlations to a manifold with a large genus *g* should then be treated as a *simplicial complex*, i.e., a connected graph of simplices. In fact, in one of the PMFG papers [130], the authors pointed out that MSTs and PMFGs should already be recognized as simplicial complexes. There is thus potential for an improved understanding of the topological structure of cross correlations in terms of simplices, but somehow Massara et al. did not pursue it further to the natural TDA conclusion.

Recently, we published a TDA paper in the *Frontier in Physics* Special Issue “From Physics to Econophysics back to Physics: Methods and Insights” [59]. In this paper, we worked out the TDA signatures for (1) coalescing spheres, (2) torus to horn torus to spindle torus to sphere, and (3) sphere to ellipsoids, and used these toy models to develop a hypothesis on market crashes corresponding to the fragmentation of a multiply connected manifold with a non-zero genus. In this hypothesis, we have the creation of holes as well as handle-breaking events that accompany fragmentations associated with market crashes. We then presented preliminary evidence confirming the existence of hole creation and handle-breaking events. In this paper, we would like to go deeper to understand how a handle breaks, or its time-reversed event, which is how two disjoint manifolds fuse with each other.

### 4.2. What Is Topological Data Analysis?

TDA is a suite of mathematical tools developed by Edelsbrunner, Zomorodian, Carlsson, and Singh to analyze the topological properties of complex data sets [55,56,57]. Built on the foundations of topology [143,144,145,146,147], group theory [148,149], linear algebra [150,151], and graph theory [152,153,154], TDA has since became a popular field in applied mathematics, and has also found many applications in data analytics [58]. For more information on the history and developments of TDA, readers can consult these review articles [155,156,157,158].

In its simplest terms, TDA is a novel way to unravel the topological features of raw data, which can be in the form of point clouds, distance matrices, networks, or digital images. To perform a TDA, we first imagine a control parameter called the *proximity parameter* or *filtration parameter*
ϵ. This is the radius of an imaginary ball centered at each of the data points, which we call 0-simplices. When we increase ϵ, the balls will grow outwards and eventually overlap with other balls. When the balls of data points *i* and *j* overlap, we draw a link between *i* and *j*, and say that the two data points now form a 1-simplex {i,j}. As ϵ increases further, there will be more overlaps, and if the k+1 data points $\left\{i_{1},i_{2},\cdots ,i_{k+1}\right\}$ are such that the balls of $i_{\alpha }$ and $i_{\beta }$ overlap, for all pairs of $\left(i_{\alpha },i_{\beta }\right)$ in the set, then we say that $\left\{i_{1},i_{2},\cdots ,i_{k+1}\right\}$ forms a *k*-simplex. The topological information contained in the data set can then be expressed in terms of the distribution of *k*-simplices, k=0,1,⋯, and how they are connected to each other into a *simplicial complex*. For different ϵ, we have different connected subsets of the simplicial complex. The *homology group**H* of a simplicial complex summarizes, in a group-theoretic way, the connectivities between *k*-simplices of different dimensions. As we analyze $H_{n}$, the *n*-dimensional subgroup of *H*, for n=0,1,⋯ over the filtration process, we will discover simplices that remain the same over a large range of ϵ, as well as those that exist fleetingly over very small ranges of ϵ. We call the former the persistent homology of the data set, and based on these we construct useful TDA metrics such as barcodes, persistent diagrams, persistent landscapes, and also persistent Betti numbers. In addition, we combine these to design other tools, such as persistence-weighted kernels, or persistent entropy, and other persistent functions. To allow readers to more easily to grasp the general idea, we show cartoons in Figure 4 to demonstrate how TDA can be applied to a data cloud.

Recently, TDA has found applications in many areas. These include computer network structures [159,160,161], computational biology [162,163,164,165,166,167,168], image analysis [169,170,171,172], vision [170], data analysis [58,173,174,175,176], shape recognition [177], and amorphous material structures [178,179]. More recently, we found the use of TDA in the reconstruction of brain functional networks [180,181], the analysis of financial markets [182,183], and haze detections [184,185]. In fact, TDA has become so much of a cottage industry that many softwares have become available for non-experts. These include Javaplex [186], Dipha [187], jHoles [188], Simpers [189], R-TDA [190], GUDHI [191], PHAT [192], Persus [193], Dinoysus [194], Ripser [195], as well as those reviewed by Otter et al. [155], and Pun et al. [158].

To the best of our knowledge, so far only very few works [182,183,196,197] had applied persistent homology and TDA to the study of trading networks, banking systems, and market crashes. The work closest to ours is that by Gidea and Katz in 2018, who treated the daily log-returns of S&P 500, DJIA, NASDAQ, and Russell 2000 as a four-dimensional data point [183]. They then slided a *w*-day time window one day at a time to create a sequence of point-cloud data sets that covered the Dotcom Crash of 2000, as well as the Global Financial Crisis of 2007–2009. The topological features they identified from the filtration process are high-order temporal correlations at various time scales. They then devised an $L_{p}$ norm that can differentiate between persistent landscapes in two time windows, revealing early warning signals preceding crashes. Building on top of this work by Gidea and Katz, as well as the econophysics literature on MSTs and PMFGs, we will report in this paper an understanding of market crashes at levels of detail never before accomplished.

### 4.3. Using TDA to Understand Market Crashes

In this subsection, we examine the cross-correlation matrices of 671 stocks in the Taiwan Stock Exchange (TWSE) in successive six-month time windows that are seven days apart, and attempt to use the toy-model results in Appendix C to understand the fusion and fission processes associated with the March 2020 crash in greater details. In particular, we would like to ask “how many of each kind of processes do we find?” and “are there combinations of more than one kind of processes?” To answer these questions, we first organize in Table 1 the Betti numbers read off at the largest filtration parameters, for time windows between 1 August 2019 and 31 March 2020. Here, we see that, over the four time windows of August 2019, we have $\beta_{0}=1$, $\beta_{1}=6.5$, and $\beta_{2}=33.75$ on average. Then, in the first two time windows of September 2019, while $\beta_{0}=1$ and $\beta_{2}=40$ remained similar to those in August 2019, $\beta_{1}$ changed from an average of $\beta_{1}=6.5$ to $\beta_{1}=1.5$. For the next three time windows, the topological changes appear to have accelerated. Using the same $\epsilon_{\max}=1.1$ over the five time windows, we found that the number of simplices increased dramatically from 10 million in the first time window of September 2019 to 85 million in the last time window of September 2019. In this last time window of September 2019, the Javaplex program failed to return any Betti numbers. It was only when we decreased the maximum filtration parameter from $\epsilon_{\max}=1.1$ to $\epsilon_{\max}=1.0$, that the number of simplices was reduced to 17 million, giving us $\beta_{0}=6$, $\beta_{1}=19$, and $\beta_{2}=17$.

To put these $\beta_{0}$ changes in the proper context, let us recall that we analyzed 671 stocks in the TWSE. When ϵ=0, none of these would be within the ϵ-ball of each other, and thus we found $\beta_{0}=671$. As ϵ increases, links start to form between stocks, and $\beta_{0}$ would decrease. After some point, the change in $\beta_{0}$ would be dominated by the gaps between clusters of stocks. If there are three such clusters, we would find $\beta_{0}=3$ over a wide range of ϵ, before it drops to 2, and then eventually to 1. This is the picture we should have in mind when we say that the persistent Betti number is $\beta_{0}=3$. However, for the last few time windows, we cannot be sure that the $\beta_{0}$ found by Javaplex are its persistent values. Physically, it is meaningful to compare persistent Betti numbers. It is also meaningful (but less so) to compare Betti numbers for a given value of ϵ. However, it is meaningless to compare Betti numbers obtained with different filtration parameters if they are not all persistent. Based on our past experience, there seems to be an analogy between the filtration parameter and the temperature of a thermodynamic system. Normally, a 10% change in the filtration parameter ϵ results in a corresponding 10% change in the number of simplices (akin to the number of arrangements whose logarithm gives us the entropy [181,198]), and hardly any changes to the Betti numbers, if we have already arrived at their persistent values. However, when the system is close to a critical point, a small change in temperature can produce a large change in the number of accessible states (analogous to simplices). To put it simply, our analysis of the Betti numbers suggests that the cross correlations in the first six time windows were more or less similar topologically, whereas for (15 September 2019, 15 March 2020) and subsequent time windows, the Betti numbers became extremely sensitive to ϵ over a broad range of ϵ, suggesting a non-trivial topological transition over the last three time windows. Another signature of this topological transition is the *persistence weakening* phenomenon that we observed in our earlier paper [59], where we found first a slow increase in the number of simplices, and then a rapid increase in the number of simplices after some threshold.

With the above in mind, let us consider the topological changes going from the second time window to the third time window, where we are confident that the Betti numbers obtained are persistent. Between these two time windows, we found that $\Delta \beta_{0}=0$, $\Delta \beta_{1}=-4$, and $\Delta \beta_{2}=+2$. Comparing these against the results of Appendix C, we realized that there were no fusions ($\Delta \beta_{0}<0$) or fissions ($\Delta \beta_{0}>0$), and therefore, none of the toy models we considered in Appendix C would be able to explain the changes to $\beta_{1}$ and $\beta_{2}$. In fact, for these first few time windows, the changes to $\beta_{1}$ appeared to be independent of changes to $\beta_{2}$, i.e., the creation/annihilation of holes seems to be independent of the creation/annihilation of voids. Some possible mechanisms for doing so are shown in Figure 5a–f. Although these time windows were still far from the crash, the picture of the market dynamics they suggest is more complex than we expected. We might need to go to higher-order Betti numbers to fully elucidate this dynamics.

In Appendix D, we showed that it is possible to have persistent Betti numbers (and thus equally meaningful pictures) at different scales. However, this makes the identification of the persistent $\beta_{0}$ more difficult, because we need to identify the filtration parameter values at which the lifetimes change most rapidly. Frequently, these are close to the largest scale, and cannot be easily seen from a full barcode (see Appendix A). To perform this multiscale analysis, we need to restrict ourselves to the longest-living barcodes, as shown in Figure 6 for the seventh of our nine time periods, i.e., (15 September 2019, 15 March 2020). In this figure, we find four persistent $\beta_{0}$ values at different scales. For the lowest of these four scales, from 0.91≤ϵ≤0.93, we have $\beta_{0}=19$. Thereafter, from 0.955≤ϵ≤097, we have $\beta_{0}=11$, and then $\beta_{0}=6$ for 0.985≤ϵ≤1.015, and $\beta_{0}=4$ for 1.02≤ϵ≤1.04). In Figure 6, the filtration parameter ends at ϵ=1.05. If we continue to increase ϵ, it is likely that we would find another persistent $\beta_{0}=2$ at a higher scale. Unlike for the seventh time period, which illustrated our ideas in Figure A6 very well, similar analyses for the eighth (22 September 2019, 22 March 2020) and ninth (1 October 2019, 31 March 2020) time periods would not yield equally convincing results, because the number of simplices at ϵ≈1 is far too large for Javaplex to handle. We also do not expect to find strongly persistent $\beta_{0}$ to emerge at the scales of $\epsilon_{\max}=0.76$ for the eighth time window, and $\epsilon_{\max}=0.65$ for the ninth time window.

To wrap up this section, we now understand that it is only meaningful to compare persistent Betti numbers or the Betti numbers at a fixed ϵ. However, we also realized from our analysis in Figure A6 that persistent Betti numbers can emerge at multiple scales, and the way to find them is to check where the lifetimes change most rapidly in the barcodes. Although we could not elucidate the persistent Betti number changes for the eighth and ninth time periods (for the March 2020 crash in the TWSE), our analyses of the first few time periods, as well as the seventh time period, are already a testament to the power of TDA. Without TDA, we would not have even guessed the roles of non-fission processes. Certainly, analyses based on the MST and PMFG would not be able to detect nucleation, rupture, and puncture events. Naturally, we need to ask whether such events are important, since these topological changes are not as drastic as fusion or fission. In any case, we must first be able to detect these events before we can evaluate how important they are relative to fusions and fissions. One hint that they might not be of negligible importance is the observed sequences of changes to $\beta_{2}$ and then to $\beta_{1}$ before $\beta_{0}$ changes. Therefore, any method that can detect $\Delta \beta_{1}$ and $\Delta \beta_{2}$ has the potential to provide early warning for fusion or fission events with $|\Delta \beta_{0}|>0$.

## 5. Going Beyond TDA: Ricci Curvature

Up to this point, we have a very detailed picture of global topological changes to the TWSE over the March 2020 crash. However, metrics such as Betti numbers cannot tell us which stocks participate in which stage of the changes. We can of course manually inspect the output of Javaplex to identify persistent simplices and then track their changes over the time windows. As can be imagined, this is extremely laborious. We would certainly like to have a metric that would automatically pick out not all persistent simplices, but those in the midst of rapid changes. It turns out that such a metric exists, after applied mathematicians recently adapted the idea of Ricci curvature to networks.

### 5.1. Ricci Curvature and Ricci Flow

To understand Ricci curvature and Ricci flow, we need to start with the *Riemannian metric*
$g_{\mu \nu }$, which allows us to specify the distance d(x,y) between any two points *x*, *y* on a surface. The Riemannian metric $g_{\mu \nu }$ is also important for the calculation of area. To explain this, let us introduce a disk B(x,r) of radius *r* centered at *x*. This is the set of all points *y* whose distance d(x,y) to *x* is less than *r*. On a Euclidean plane, the area |B(x,r)| of B(x,r) would be $\pi r^{2}$. On a Riemannian surface, however, this area can deviate from $\pi r^{2}$. To understand this deviation, let us imagine a disk on the surface of a sphere. Through elementary calculus, we can show that the area of such a disk is a little less than $\pi r^{2}$, and we can understand this deficit as due to the *scalar curvature*
(3)$$R\left(x\right):=\lim_{r\to 0}\frac{\pi r^{2}-\left|B(x,r)\right|}{\pi r^{4}/24}$$
on the surface of the sphere. One of the main disadvantages of using the scalar curvature is that we do not know whether the curvatures along different directions are the same, or different. Therefore, we extend the notion of the scalar curvature to directional curvatures by defining the *Ricci curvature* as
(4)$$Ric\left(x\right)\left(\nu ,\nu \right):=\lim_{r\to 0}\lim_{\theta \to 0}\frac{\frac{1}{2}\theta r^{2}-\left|A(x,r,\theta ,\nu )\right|}{\theta r^{4}/24},$$
for an angular sector A(x,r,θ,ν) inside a small disk B(x,r), which has a small angular aperture θ (measured in radians) centered around some direction ν (a unit vector) emanating from *x*. Here, |A(x,r,θ,ν)| is the area of the small angular sector, and Ric(x)(ν,ν) is the inner product of the Ricci curvature tensor along the ν direction. If Ric(x) has the same value for all ν, we say that the curvature of the surface is isotropic at *x*. Otherwise, the curvature at *x* is anisotropic.

The definition in Equation (Equation 4) allows the Ricci curvature to be computed intrinsically, i.e., without embedding the surface in a higher-dimensional space. This property is important when we generalize the Ricci curvature to networks. Going back to surface of a sphere, we will find that Ric(x) has the same positive value at every *x*, and for every direction ν. Therefore, the Ricci curvature on the surface of a sphere is not only isotropic, it is also positive. For a planar surface, the Ricci curvature is also isotropic at all points, but its value is zero. For an arbitrary two-dimensional surface, the Ricci curvature at a given point will vary from some maximum value to some minimum value. These two values are called the *principal curvatures* of the surface at the given point. In general, for an *n*-dimensional surface, the Ricci curvature will vary between *n* principal curvatures, each of which can be positive, negative, or zero. A highly readable explanation can be found in Terence Tao’s blog [199].

The Ricci curvature plays an important role in Einstein’s theory of general relativity [200]. Even though Einstein worked through the Riemann curvature tensor $R_{\mu \sigma \nu }^{\rho }$, to get to the Ricci curvature $R_{\mu \nu }=\sum_{\rho ,\sigma }R_{\mu \sigma \nu }^{\rho }$ and the scalar curvature $R=\sum_{\mu ,\nu }R_{\mu \nu }$, we note that the Riemann curvature tensor is coordinate-dependent, while the Ricci curvature and scalar curvature are both coordinate-independent. It therefore makes perfect sense that only coordinate-independent quantities can enter Einstein’s field equations. Another application of the Ricci curvature is its use to measure the growth of volumes of distance balls, transportation distances between balls, divergence of geodesics, and meeting probabilities in coupled random walks [201].

Due to its intrinsic character, the Ricci curvature is also the central concept behind the theory of *Ricci flow*,
(5)$$\frac{d}{dt}g=-2Ric,$$
which is the mathematical theory that describes how manifolds deform. Informally, Ricci flow is the process of stretching the Riemannian metric *g* (increasing distance between points) in directions of negative Ricci curvature, and contracting *g* (decreasing distance between points) in directions of positive Ricci curvature. The stronger the curvature, the faster the stretching or contracting of the metric. In principle, one can use this equation to perform Ricci flow on a manifold for as long a period of time as one wished. In practice, however, it is possible for a manifold to develop singularities (where the curvature becomes infinite) during the Ricci flow. In three dimensions, many complicated singularities are possible. For instance, one can have a neck pinch, in which a cylinder-like “neck” of the manifold shrinks under Ricci flow, until at one or more places along the neck, the cylinder has tapered down to a point.

In pure mathematics, the theory of Ricci flow was instrumental in the proof of the Poincare conjecture (see Appendix E). So how does Ricci flow connect to what we care about in complex systems, or econophysics in particular? Conventionally, before one looks into the dynamics of a complex system, the first parsimonious step will always be to examine only the backbone (the “topology”) of the dynamics. From this perspective, MSTs, PMFGs, graphs, networks, manifolds, or simplicial complexes are different constructs to inform us what this backbone is like. After constructing the backbone, and making sure that it is roughly correct, we then add the “geometry” of the dynamics in as a natural second step, and a natural and coordinate-independent way to quantify this would be to use the Ricci curvature. Therefore, it is important not to go directly into the geometry, before getting a perspective on the topological panorama, because the same curvature value can often mean different things when they are put onto different topologies. For example, the *n*-sphere and the *n*-torus are topologically different manifolds, but they could still have similiar average curvatures. Therefore, the use of curvature alone cannot distinguish them. This is also why the correct procedure should always work on the topologies first, before putting the curvatures back, to acquire the correct geometrical information.

From the viewpoint of theorists, the use of differentiable manifolds to describe complex system dynamics is rigorous. In real-world problems, however, manifold constructs are difficult to implement, due to computational limitations. Hence, our plan B often involves the coarse-graining of smooth manifolds. The way this works is to first collect real-world time series cross-section data and calculate their correlation matrices, before visualizing them in terms of networks or simplicial complexes to extract their topological characteristics. From this perspective, we are interested in the breaking of bridges, or the fusion of clusters. For differentiable manifolds, the different types of singularities that can be encountered in two-dimensional Ricci flow have been completely worked out [202], and partially so for Ricci flow in three dimensions [203,204]. For higher dimensions, these are still poorly understood [205]. For networks and simplicial complexes, we need to start with discrete versions of the Ricci curvature. These are the Ollivier-Ricci curvature (ORC) [206,207] and the Forman-Ricci curvature (FRC) [208,209]. The former is applied to networks, whereas the latter is devised for simplicial complexes. It has been found that the ORC is “related to” various graph invariants, ranging from local measures, such as the node degree and clustering coefficient, to global measures, such as betweenness centrality and network connectivity [210]. Thus far, ORC has been used to broadly investigate properties of the internet [210], gene expression networks related to cancer [211], and the structural connectivity of an animal brain [212], as well as to assist in specific tasks such as community detection [213,214], the measurement of market fragility, and the estimation of systemic risk [215]. In this work, we focus on using the ORC, and defer the application of the FRC to future works.

To define the ORC in mathematical terms, we start with an unweighted graph G=(V,E) with vertex set $V={\left\{x_{i}\right\}}_{i=1,\cdots ,n}$ and edge set $E={\left\{e_{k}\left(x_{i_{k}},x_{j_{k}}\right)\right\}}_{k=1,\cdots ,m;i_{k},j_{k}\in V}$, where *n* is the total number of vertices and *m* is the total number of edges. Let $N_{x}$ be the neighborhood of a vertex x∈V. To introduce a curvature measure on a graph, Ollivier associated curvature with transport processes, much like the original concept of curvature being related to the parallel transport of one tangent vector along another. On a graph, the natural transport process to consider is a random walk, and the natural analog of parallel transport is how the hopping probabilities $\mu_{x}\left(x^{\prime }\right)$ from a vertex x∈V to its neighbors $x^{\prime }\in N_{x}$ change to the hopping probabilities $\mu_{y}\left(y^{\prime }\right)$ from a vertex y∈V to its neighbors $y^{\prime }\in N_{y}$ as we move the geodesic distance d(x,y) from *x* to *y*. This change can be quantified by the *first Wasserstein distance*, also known as the *earth mover distance*
(6)$$W_{1}\left(\mu_{x},\mu_{y}\right)=\inf\sum_{x^{\prime }\in N_{x}}\sum_{y^{\prime }\in N_{y}}d\left(x^{\prime },y^{\prime }\right)\xi^{xy}\left(x^{\prime },y^{\prime }\right)$$
where inf is the infimum, and $\xi^{xy}\left(x^{\prime },y^{\prime }\right)$ represent the amount of “mass” moved from $x^{\prime }$ to $y^{\prime }$, so that, after all movements, the hopping probabilities change from $\mu_{x}$ to $\mu_{y}$. In the original paper by Ollivier, and others after him, the hopping probabilities $\mu_{x}$ are defined as
(7)$$\mu_{x}\left(x^{\prime }\right)=\left\{\begin{array}{cc} \frac{1}{|N_{x}|}, & \mathrm{if}\;x^{\prime }\in N_{x};\\ 0, & \mathrm{otherwise}, \end{array}\right.$$
where $|N_{x}|$ is the total number of neighbors in $N_{x}$. In the eighth example of his 2009 paper [207], Ollivier considered a *lazy random walk*, and used a modified set of hopping probabilities
(8)$$\mu_{x}\left(x^{\prime }\right)=\left\{\begin{array}{cc} \frac{1}{2}, & \mathrm{if}\;x^{\prime }=x;\\ \frac{1}{2|N_{x}|}, & \mathrm{if}\;x^{\prime }\in N_{x};\\ 0, & \mathrm{otherwise}. \end{array}\right.$$

This modification is useful, because in general, random walk on a graph *G* does not always lead to a stationary probability distribution, whereas a lazy random walk always do. Finally, in terms of $W_{1}\left(\mu_{x},\mu_{y}\right)$, the ORC can be defined as
(9)$$\mathrm{ORC}\left(x,y\right):=1-\frac{W_{1}\left(\mu_{x},\mu_{y}\right)}{d(x,y)}.$$ This can be obtained using a linear programming procedure to optimize $W_{1}\left(\mu_{x},\mu_{y}\right)$, as shown in Appendix F. In this Appendix, we computed $W_{1}\left(\mu_{x},\mu_{y}\right)$ for two arbitrary nodes *x* and *y* on the network *G*, but in Equation (Equation 9), as part of the definition for ORC(x,y), we compute $W_{1}\left(\mu_{x_{i_{k}}},\mu_{x_{j_{k}}}\right)$ only for edges $e_{k}\left(x_{i_{k}},x_{j_{k}}\right)\in E$. For the more common graphs, i.e., tree graphs, grid graphs, complete graphs, or bipartite graphs, ORC(x,y) can be evaluated in simple mathematical forms.

### 5.2. Ollivier-Ricci Curvature Analysis of TWSE

After confirming using the toy model in Appendix G the utility of negative ORCs to identify neck regions, we turn our attention to the TWSE March 2020 crash. The neck regions in the simplicial complexes across this market crash should also be thin and weakly connected parts that are most likely to be associated with rapid changes. Since the ORC computation requires a graph as input, we have to produce one starting from the Pearson cross correlations. Therefore, in the first part of this subsection, we limited ourselves to one time period (1 October 2019, 31 March 2020), and explore different ways to create the input graph.

Naively, we can create a complete network in which all links are present, but with different weights. However, such a network will always look like a fur ball when visualized, making it difficult for us to discern the various neck regions. Therefore, the first thing we tried is *threshold filtering*, i.e., we draw a link between stocks *i* and *j* if $C_{ij}>C_{0}$. The Python function that computes the ORC can accept as input disconnected graphs, but we adjusted $C_{0}$ until we obtain a fully connected graph. Unfortunately, for this (1 October 2019, 31 March 2020) time period, the fully connected graph obtained for the TWSE has 166,831 links. After visualization it still looks like a fur ball, impeding our investigations of topological changes in the network.

The next thing we tried is the *minimal spanning tree*, which can be constructed using the Kruskal algorithm shown in Figure 3a. Compared to a fur ball, the MST is more informative, especially when we used the *force atlas layout* [216]. In this layout, shown in Figure 7, nodes that are connected by short links have strong Pearson cross correlations, whereas those that are connected by long links have weak Pearson cross correlations. This geometrical feature of the layout allows us to discern clusters of strongly correlated nodes, separated from each other by weak correlations with bridging nodes. However, in the MST only, N−1 links are retained for *N* nodes. These are very few, so we checked how many important links were rejected by the MST in the nine periods, using two measures of importance: (1) correlations larger than the minimum correlation incorporated into the MST, and (2) correlations larger than the minimum correlation associated with the hub of the MST. These are shown in Table 2. Indeed, a large number of cross correlations larger than (1) were rejected in all nine time periods, and especially in the last two time periods. However, the importance measure (1) may be too strict, since we know that for all nodes to be connected in the MST, we frequently have to incorporate weak cross correlations. Based on importance measure (2), which is the correlation level set by the hub, the number of rejected cross correlations is significantly fewer, except during the fifth, seventh, and ninth time periods.

Therefore, the last filtering we tried is the *planar maximally filtered graph* (PMFG), adapting the Python example in https://gmarti.gitlab.io/networks/2018/06/03/pmfg-algorithm.html, accessed on 5 May 2021. As we have described in Section 3, this information filtering method was first proposed by Tumminello et al. [50]. We should add that in recent implementations, the Boyer–Myrvold planarity test [217] has replaced the Kuratowski theorem [153] for checking that the graph remains planar at different stages. The resulting PMFG is shown in Figure 8i. By allowing cycles, clusters are more compact in the PMFG. Additionally, 3(N−2) links were kept. This is an intermediate number that is still easy to visualize, and contains more of the important cross correlations. In particular, we observed that the cluster at the bottom of the visualization is connected to the rest of the network through two necks (instead of one). However, if we use the same two measures of link importance as for the MST, we see in Table 3 even more important cross correlations rejected in the PMFGs.

After deciding to use the PMFG visualization across all time periods, we tried to identify neck regions that persisted over several time periods to better understand how the market crash proceeded. Therefore, we used the final layout of the first time period as the initial layout of the second time period, the final layout of the second time period as the initial layout of the third time period, and so on and so forth. We had hoped that the PMFGs for successive periods would be sufficiently similar that we could identify features across them. Unfortunately, as we can see from Figure 8, this is not the case, even when we reduced the number of iterations to 100 for the force atlas layout algorithm.

Due to this problem, we abandoned our original ambitious plan to automatically identify all neck regions and their changes. Instead, we manually analyzed the neck region that changed the most dramatically over the market crash. To begin, we first plotted in Figure 9 the number of links with strongly negative ORCs (<−0.5) over the time periods. As we can see, the number of such links increased as we approached the March 2020 market crash, but the number also increased in the third and fourth periods before falling back to levels close to the first and second periods. By checking the number of links with ORC <−0.45 and the number of links with ORC < −0.55, we see that these features are robust and associated with strongly negative ORCs. It appears therefore that a rising number of links with strongly negative ORCs is also an early warning indicator of a market crash. This was first observed by Sandhu et al. [215].

After inspecting the lists of links with ORC<−0.5, we focused on two links, (176, 193) and (176, 393), which (1) appeared frequently in the PMFGs across the nine time periods, and (2) had consistently negative curvatures. These three stocks are: (176) Tung Thih Electronic Co., Ltd., Taoyuan City, Taiwan (3552.TWO); (193) C-Tech United Corp., New Taipei City, Taiwan (3625.TWO); and (393) Taiwan Semiconductor Co., Ltd., New Taipei City, Taiwan (5425.TWO). (176) Tung Thih Electronic Co., Ltd. is a large company in the Auto Parts industry, with market capitalization 15.11 billion TWD, whereas (193) C-Tech United Corp. is a medium-size company in the Electrical Equipments & Parts industry, with market capitalization of 1.45 billion TWD. The last company, (393) Taiwan Semiconductor Co., Ltd., is another large company in the Semiconductors industry, with a market capitalization of 10.5 billion TWD. To put the sizes of these companies into the proper perspective, we compare them against TSMC (2330.TWO), the largest chip maker in the world and one of the largest companies in Taiwan, with a market capitalization of 14.73 trillion TWD. The ORCs of these links over the nine periods are shown in Table 4. Over the period of study, there are no PMFG links between 193 and 393.

From Table 4 we see that ORC(176,393) is less strongly negative, and change more slowly than ORC(176,193). Since the link (176,193) did not appear in the last time period, we suspect that the cluster associated with 193 has completely broken off from the cluster associated with 176. More importantly, comparing Table 4 and Figure 9, we see that the appearance of (176,193) in the PMFG coincided with the periods when the number of links with strongly negative curvature was increasing. This suggests that the link (176,193) might have formed in the third time period, broke off in the fifth time period and thereafter reformed in the sixth time period, before finally breaking up in the last time period. Such a sequence of events would surely be interesting to elucidate, but a detailed story might be better suited for a future study that we hope to do using the Generalized Forman-Ricci curvature [218] to more closely track how these fusions and fissions unfold. To wrap this paper up, let us visualize the clusters that these three nodes participate in over the last three time periods.

In network science, in addition to global layout algorithms for visualizing entire networks, we also find ego-centric visualizations centered on a node that are of interest. In Figure 10, we chose 176, 193, and 393 to be the three centers we would like to visualize around. Then, we included all nodes in the immediate neighborhoods of 176, 193, and 393, and colored the links they have with 176, 193, and 393 red if they have ORC<−0.2 (strongly negative), green if −0.2≤ORC≤0.2 (roughly zero), and blue if ORC>0.2 (strongly positive). Next, we drew only green and blue links between the neighbors of 176, 193, and 393, omitting red links between them. Finally, we colored simplices bound by green or blue links yellow. In this way, we keep the number of nodes and number of links to be visualized in Figure 10 to manageable numbers.

From this figure, we see in the seventh time period (15 September 2019–15 March 2020) that 176, 193, and 393 lied at the peripheries of the clusters they belong to respectively. We know that these nodes were at the peripheries of their respective clusters, because in their ego-centric visualizations, they would be surrounded by mostly green or blue links if they were part of the cores of their clusters. In this time period, 176 and 193 were connected directly through a red link, but the two of them were connected to 393 through 318 (Asia Electronic Material Co., Ltd., Zhubei, Taiwan (4939.TWO), Electronic Components). In the eighth time period (22 September 2019–22 March 2020), we see that the clusters containing 176 and 193 had merged, even though the two nodes were still at the fringe of this merged cluster, and still connected by a red link. 193 remained unlinked to 393, but 176 had “robbed” 318 from 393, but was now directly linked to 393 through a red link, as well as via 389 (AVY Precision Technology Inc., Taipei City, Taiwan (5392.TWO), Electronic Components) and 468 (Netronix Inc., Hsinchu City, Taiwan (6143.TWO), Communication Equipment). 176 also had other red links with the cluster 393 belonged to. Finally, in the last time period (1 October 2019–31 March 2020), we see that the cluster 193 belonged to had completely broken off from 176 (within the PMFG visualization for the entire network). Interestingly, 193 retained its link to 607 (Firich Enterprises Co., Ltd., New Taipei City, Taiwan (8076.TWO), Computer Hardware) from the eighth time period, and regained its connection to 318, at the same time made a new connection to 196 (Newmax Technology Co., Ltd., Taichung, Taiwan (3630.TWO), Electronic Components). Going from the eighth time period to the ninth time period, the biggest change (related to 176, 193, and 393) would be the clusters associated with 176 and 393 merging into a giant cluster. In this giant cluster, 176 and 393 were still peripheral nodes, but there was now a green link between them. In addition, 176 and 393 were also connected by green links through 178 (eGalax_eMPIA Technology Inc., Taipei City, Taiwan (3556.TWO), Semiconductors) and 190 (AimCore Technology Co., Ltd., Hsinchu City, Taiwan (3615.TWO), Electronic Components). In the eighth time period, 178 was connected to 176 by a green link, but not connected to 393, whereas 190 was connected to 393 by a green link, and to 176 by a red link in this time period.

To summarize, changes to the neck regions between 176, 193, and 393 appeared to be very sudden, even when we slid the time window by only seven days. This suggests the need to slide the time window through a smaller time step, to properly track changes to the network of stocks. However, to use such small time steps meaningfully, we will have to use intra-day time series data, instead of the daily data that we used in this paper. Furthermore, none of the fusions and fissions in Figure 10 resemble the simple toy models A or C (single neck, with fixed or varying dimensionality) described in the Appendix, even though it appears that these do start at the periheries of clusters. However, some aspects (multiple distant necks) of these events are similar to what happens in toy models B or D. In this sense, we are starting to understand why re-organizations of cross correlations in the financial market lead frequently to topological features such as voids.

## 6. Conclusions

Over the past 20 years, state-of-the-art information filtering methods such as the MST and the PMFG have revolutionized the field of econophysics, and also made contributions to other closely related disciplines. In this paper, we suggested two related directions to extend this information-filtering paradigm. The first is through topological data analysis (TDA), and the second is through the calculation of Ollivier-Ricci curvature. The former improves our understanding of the topological backbones of financial networks, whereas the latter puts the geometrical information back onto the topological backbones.

In the TDA, we explored four toy models of fusions, namely (1) the merging of two ellipsoidal surfaces, (2) the merging of two biconvex surfaces, (3) the merging of two anisotropic ellipsoidal surfaces through a sequence of higher-dimensional connections, and finally (4) the merging of two random irregular surfaces. By applying the insights extracted from this exploration to a recent crash in the TWSE, we found the number of simplices increasing slowly with increasing filtration parameter ϵ half a year before the market crash, and rapidly with increasing ϵ close to the crash. This suggests a non-trivial topological transition accompanied the market crash. However, we found that the four fusion/fission models proposed were not able to fully explain the topological changes, and additional processes (cavitation, annihilation, rupture, healing, and puncture) that do not involve fusion or fission, were needed to explain the changes in Betti numbers.

Moving beyond TDA, we used the Ollivier-Ricci curvature to quantify the distribution of curvatures in PMFGs constructed from the correlation matrices of the TWSE. We explained that positive ORCs correspond to stock components deep within a cluster, whereas negative ORCs pinpointed the neck (bridge) regions that connect distinct clusters. When we examined the PMFGs for nine periods between August 2019 and March 2020, we found dramatic topological changes between successive periods. This prevented us from systematically identifying all topological changes that were specifically associated with neck regions in the PMFGs. Instead, we look only at two neck regions—associated with the links (176, 193) and (176, 393)—that featured prominently during this period. These three nodes are: (176) Tung Thih Electronic Co., Ltd, (193) C-Tech United Corp., and (393) Taiwan Semiconductor Co., Ltd. During the last time period, (176, 193) was no longer found in the PMFG, while the curvature of (173, 393) became nearly zero. Using ego-network visualizations of these three nodes and selective visualization of links between them, we saw that all three nodes lie on the peripheries of the clusters they belonged to. In the seventh time period, all three clusters were distinct. In the eighth time period, the cluster containing 176 merged with the cluster containing 193. Finally, in the ninth time period, this cluster broke up into a small cluster containing 193, while the larger cluster containing 176 proceeded to merge with the cluster containing 393.

## Figures and Tables

**Figure 1 entropy-23-01211-f001:**
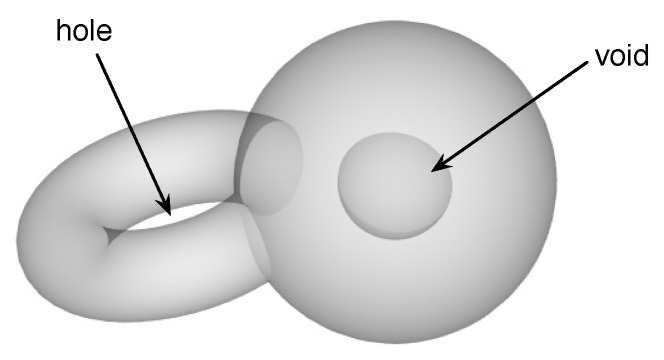
A manifold with a “hole” as well as a “void”.

**Figure 2 entropy-23-01211-f002:**
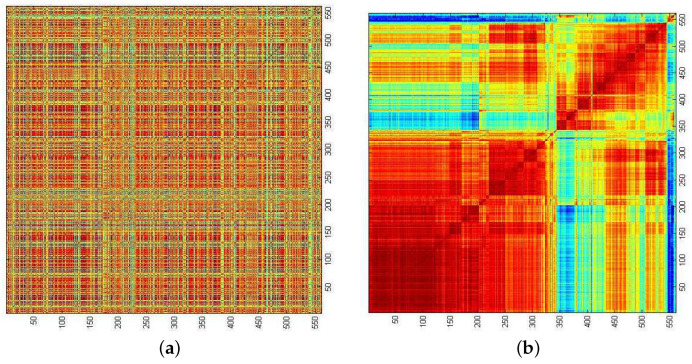
(**a**) The cross-correlation matrix for 561 stocks in the SGX from January 2008 to December 2009. In this figure, red correlations are strongly positive, blue correlations are strongly negative, while green correlations are close to zero. No structures can be discerned in this figure, because the stocks are arranged in alphabetical order. (**b**) After reordering the rows and columns of the cross-correlation matrix, we found strong correlations organized into diagonal blocks, with weaker correlations between them. Material from: Teh et al., Cluster fusion-fission dynamics in the Singapore stock exchange, Euro. Phys. J. B, published 2015 [67], Springer Nature Switzerland AG.

**Figure 3 entropy-23-01211-f003:**
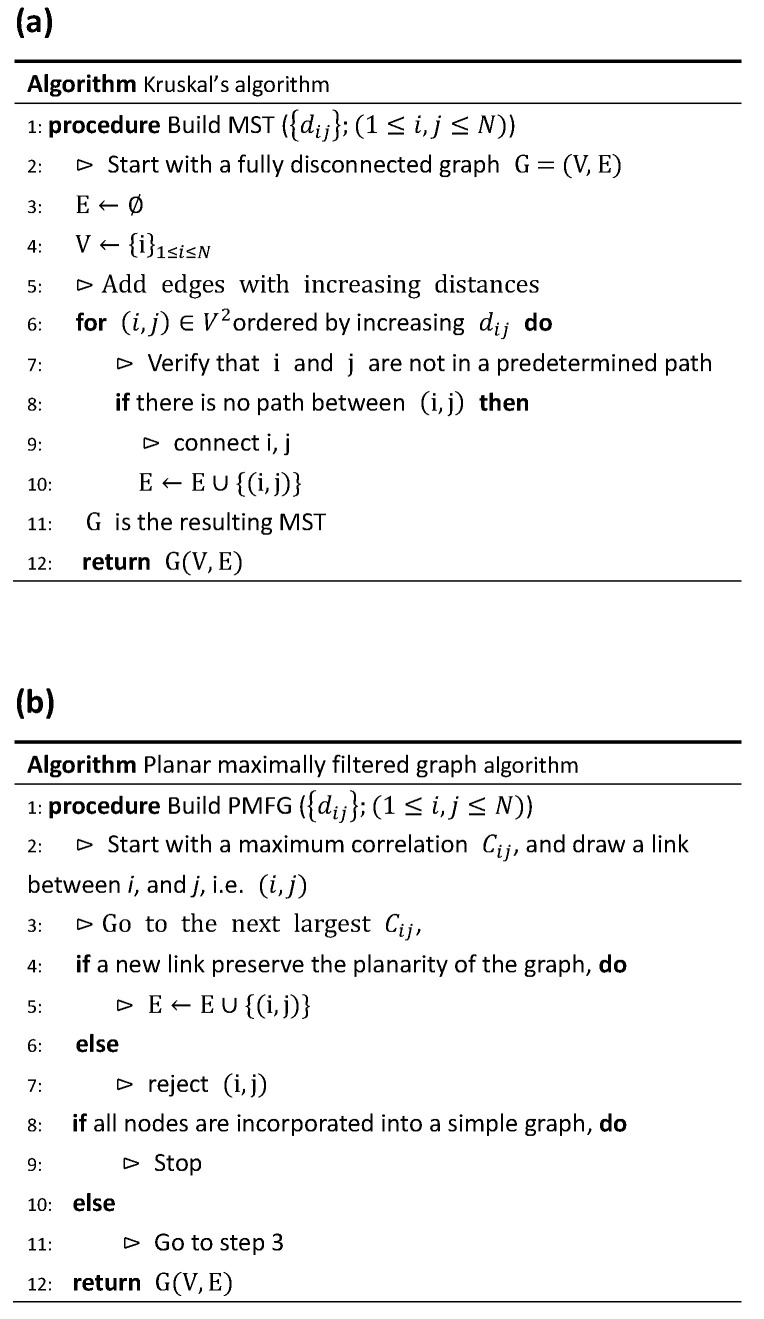
Pseudo codes for (**a**) minimal spanning tree and (**b**) a planar maximally filtered graph.

**Figure 4 entropy-23-01211-f004:**
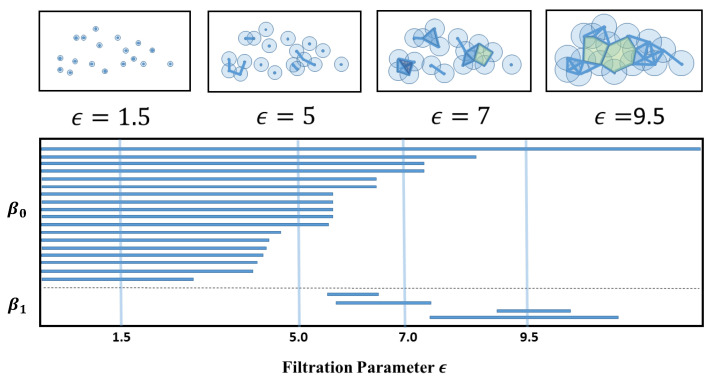
In the top row, we show a schematic diagram showing a data cloud and how the filtration process results in various overlapping outcomes for balls of different proximity parameters ϵ. In the bottom row, we show the barcodes obtained by scanning through the full range of ϵ. In this figure, we partition the barcodes into those for 0-dim simplices, which we indicate using the Betti number $\beta_{0}$, and those for 1-dim simplices, which we indicate using the Betti number $\beta_{1}$. In a barcode diagram, the barcode for a 0-dim simplex (a node) always starts at $\epsilon_{1}=0$, since all points are present at the start of the filtration process. The barcode of a 0-dim simplex ends at $\epsilon_{2}>\epsilon_{1}$, when the point is incorporated into a higher-dimensional simplex. In contrast, the barcode of a 1-dim simplex (a link) starts at $\epsilon_{1}>0$, when two balls of radius $\epsilon_{1}$ touch. The barcode of this 1-dim simplex then ends at $\epsilon_{2}>\epsilon_{1}$, when a third ball with radius $\epsilon_{2}$ (which may be part of another simplex) touches the first two. At the values of ϵ shown in the top row, we can also see $\beta_{0}$ going from 18→11→4→1, and $\beta_{1}$ going from 0→0→1→2, respectively.

**Figure 5 entropy-23-01211-f005:**
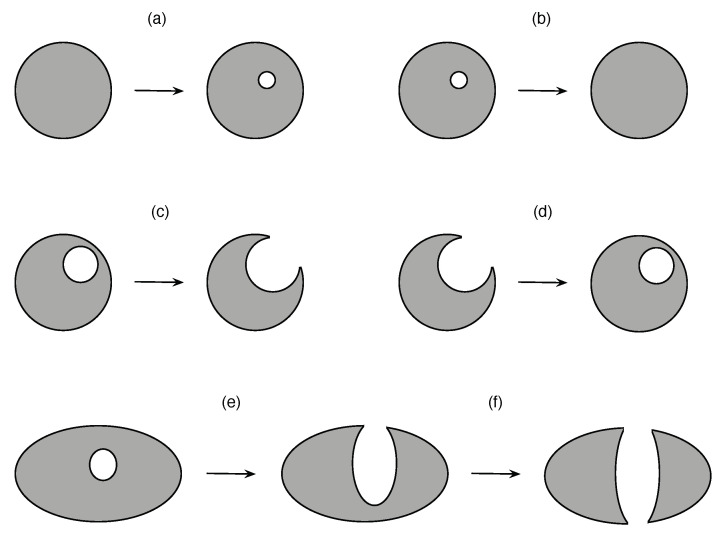
Topological changes that does not involve fusion or fission: (**a**) *cavitation*, in which a void forms within a manifold, (**b**) *annihilation*, in which a void within a manifold disappears, (**c**) *rupture*, in which a void breaks through the surface of the manifold, (**d**) *healing*, in which the surface of the manifold closes over a cavity to form a void, (**e**) another example of rupture, with the growing cavity proceeding to (**f**) *puncture* the manifold, forming a hole. The Betti number signatures of these changes are: (**a**) $\Delta \beta_{2}=+1$, (**b**) $\Delta \beta_{2}=-1$, (**c**) $\Delta \beta_{2}=-1$, (**d**) $\Delta \beta_{2}=+1$, (**e**) $\Delta \beta_{2}=-1$, and (**f**) $\Delta \beta_{1}=+1$.

**Figure 6 entropy-23-01211-f006:**
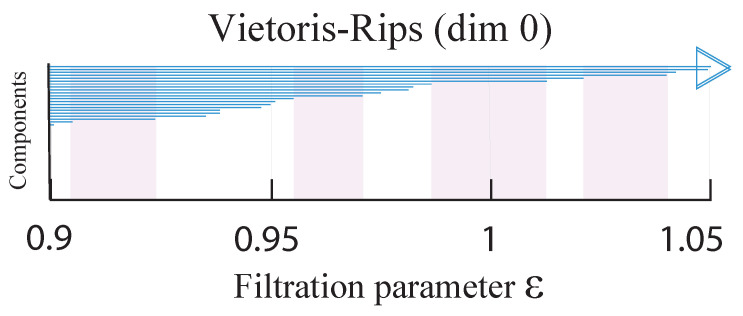
Visualization of the $H_{0}$ barcodes of TWSE in the seventh period (15 September 2019, 15 March 2020). We restrict our attention to 0.9≤ϵ≤1.05, so that we can inspect the finer details. In this figure, the persistent $\beta_{0}$ are highlighted as the pink shaded regions.

**Figure 7 entropy-23-01211-f007:**
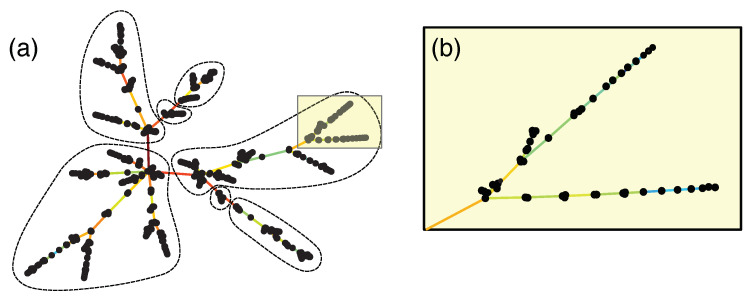
(**a**) The minimal spanning tree of 671 stocks on the TWSE, computed from their Pearson cross correlations between 1 October 2019 and 31 March 2020. In this figure, the black nodes represent stocks, while the colored links represent the most important cross correlations between stocks discovered using the Kruskal algorithm. If a link is red, it has negative ORC, whereas if a link is blue, it has positive ORC. We also sketched the seven clusters in the minimal spanning tree, using links with the most negative ORCs as a guide. (**b**) Enlarging the highlighted region in the minimal spanning tree shown in (**a**), we find that the links between closely spaced nodes have positive ORCs (and thus are shown in blue).

**Figure 8 entropy-23-01211-f008:**
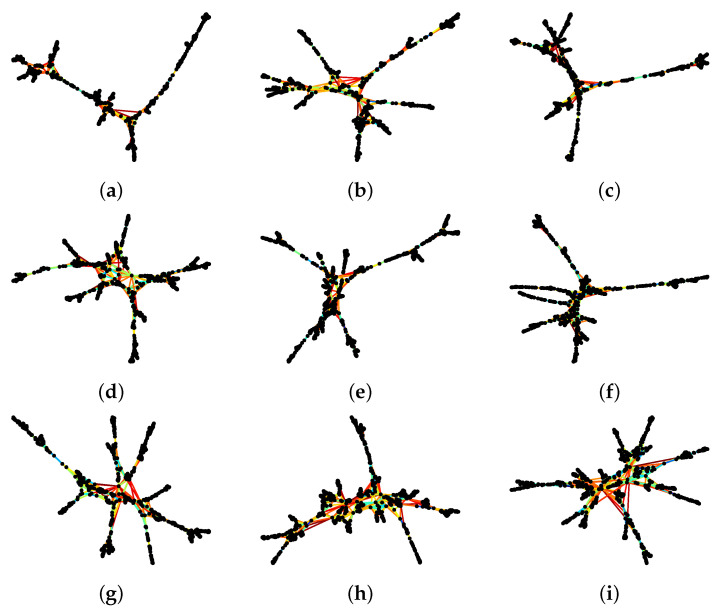
Sequence of PMFGs of 671 stocks on the TWSE, computed from their Pearson cross correlations for the time periods (**a**) 1 August 2019–31 January 2020, (**b**) 8 August 2019–8 February 2020, (**c**) 15 August 2019–15 February 2020, (**d**) 22 August 2019–22 February 2020, (**e**) 1 September 2019–1 March 2020, (**f**) 8 September 2019–8 March 2020, (**g**) 15 September 2019–15 March 2020, (**h**) 22 September 2019–22 March 2020, (**i**) 1 October 2019–31 March 2020. In this figure, the black nodes represent stocks, while the colored links represent the most important cross correlations between stocks discovered using the Kruskal algorithm. The links are colored according to their ORCs, with red being negative, green being approximately zero, and blue being positive.

**Figure 9 entropy-23-01211-f009:**
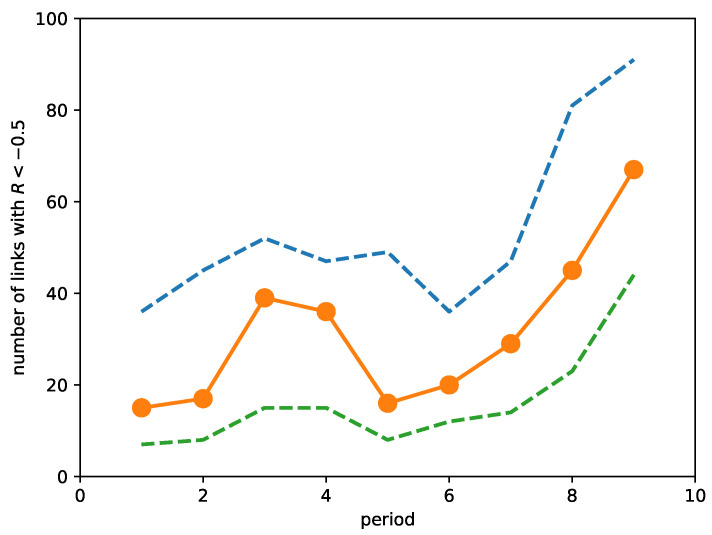
Number of links with ORC<−0.5 over the different time periods (shown in orange): (1) 1 August 2019–31 January 2020, (2) 8 August 2019–8 February 2020, (3) 15 August 2019–15 February 2020, (4) 22 August 2019–22 February 2020, (5) 1 September 2019–1 March 2020, (6) 8 September 2019–8 March 2020, (7) 15 September 2019–15 March 2020, (8) 22 September 2019–22 March 2020, (9) 1 October 2019–31 March 2020. Additionally, the number of links with ORC<−0.45 (blue dashed lines) and the number of links with ORC<−0.55 (green dashed lines) are also shown.

**Figure 10 entropy-23-01211-f010:**
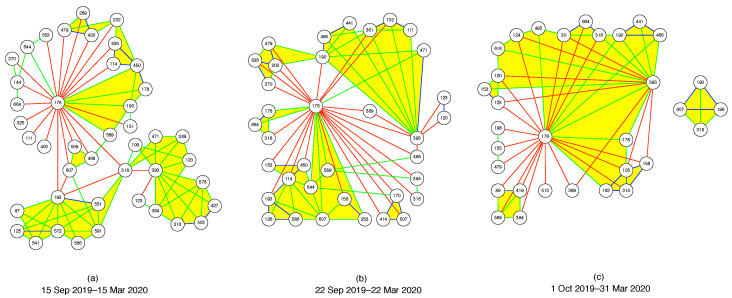
Rough sketch of the fission sequence in the TWSE from time window (**a**) 15 September 2019–15 March 2020 to time window (**b**) 22 September 2019–22 March 2020 to time window (**c**) 1 October 2019–31 March 2020. In this figure, we include the nodes 176, 193, and 393 for all three time windows. In each time window, we include all the nearest neighbors of 176, 193, and 393. We show all the links between these nodes with 176, 193, and 393, and color them red if their ORC<−0.2, green if their −0.2≤ORC≤+0.2, and blue if their ORC>+0.2. Finally, we show all green and blue links between these nearest-neighbor nodes, and color all simplices bound by green or blue links, to help visualize the clusters in the neighborhoods of 176, 193, and 393. Note that the members of these clusters are dynamic, suggesting strong mixing of cross correlations in the TWSE.

**Table 1 entropy-23-01211-t001:** The calculated Betti numbers up to k=2, total links, $\epsilon_{\max}$, and total number of simplices for TWSE during 1 August 2019 to 31 March 2021, which covers the COVID-19 crash with a sliding window of seven days.

Date	$\beta_{0}$	$\beta_{1}$	$\beta_{2}$	Links	$\epsilon_{\max}$	Simplices
(1 August 2019–31 January 2020)	1	6	23	27,675	1.1	3,444,963
(8 August 2019–8 February 2020)	1	9	31	37,696	1.1	15,194,973
(15 August 2019–15 February 2020)	1	5	33	43,708	1.1	31,321,288
(22 August 2019–22 February 2020)	1	6	48	46,507	1.1	41,178,428
(1 September 2019–01 March 2020)	1	2	40	46,944	1.1	42,079,525
(7 September 2019–8 March 2020)	1	1	40	47,482	1.1	44,068,045
(15 September 2019–15 March 2020)	2	8	19	58,201	1.1	39,877,266
(22 September 2019–22 March 2020)	65	17	1	36,504	0.75	40,871,885
(1 October 2019–31 March 2020)	91	8	1	37,640	0.65	57,119,884

**Table 2 entropy-23-01211-t002:** In this table on the nine time periods of the TWSE, we show how many stronger cross correlations were rejected in favour of weaker ones, because they would lead to the inclusion of cycles in the MSTs.

Period	1	2	3	4	5	6	7	8	9
$C_{\min}$	0.442	0.429	0.420	0.416	0.444	0.436	0.429	0.428	0.368
Links Rejected	40,519	61,601	76,327	83,437	72,813	76,895	98,935	251,601	333,363
$C_{\min}^{\left(hub\right)}$	0.707	0.781	0.689	0.823	0.545	0.729	0.697	0.912	0.730
Links Rejected	2357	977	7971	661	38,811	5607	11,389	977	121,433

**Table 3 entropy-23-01211-t003:** In this table on the nine time periods of the TWSE, we show that many stronger cross correlations were rejected in favour of weaker ones, because they would lead to the loss of planarity in the PMFGs.

Period	1	2	3	4	5	6	7	8	9
$C_{\min}$	0.018	0.103	0.097	0.012	−0.209	−0.044	−0.044	−0.079	−0.124
Links Rejected	245,846	224,312	237,712	286,892	380,440	313,430	340,352	415,946	429,102
$C_{\min}^{\left(hub\right)}$	0.407	0.401	0.459	0.448	0.545	0.597	0.454	0.600	0.705
Links Rejected	49,414	70,804	59,990	69,216	37,290	24,406	86,074	149,254	103,006

**Table 4 entropy-23-01211-t004:** Ollivier-Ricci curvatures of the links (176,193) and (176,393) in the PMFGs over the nine time periods. If the curvature value is left blank, the two nodes are not connected in the PMFG.

Period	ORC (176, 193)	ORC (176, 393)
1 August 2019–31 January 2020		
8 August 2019–8 February 2020		
15 August 2019–15 February 2020	−0.61	−0.53
22 August 2019–22 February 2020	−0.64	−0.41
1 September 2019–1 March 2020		−0.39
8 September 2019–8 March 2020	−0.59	−0.39
15 September 2019–15 March 2020	−0.55	
22 September 2019–22 March 2020	−0.37	−0.28
1 October 2019–31 March 2020		−0.07

## Data Availability

All Python and Matlab scripts are provided, along with instructions on how to use them. This will download the raw data from Yahoo! Finance and perform the necessary computations to give the final results. See the links listed in Appendix A.

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
