# Peer review of "Understanding Changes in the Topology and Geometry of Financial Market Correlations during a Market Crash"

_entropy, 2021, doi:10.3390/e23091211_

Round 1

Reviewer 1 Report

The paper provides a nice review of the network filtering methods used in econophysics and it introduces a novel method based on Ricci curvature known from general relativity. It nicely incorporates concepts from algebraic topology as Betti numbers. I really appreciate the nice description of how Bethe numbers change during various topological changes of various manifolds. The paper is really nice to read, very illustrative, and with interesting results. Therefore, I suggest accepting the paper and if possible, I would speak for further promotion of the paper. 

Reviewer 2 Report

The paper investigates new approaches for applying information-filtering methods in econophysics. The proposed methods are interesting and may bring new insights in understanding the dynamics of stock markets, however, the presentation should be improved making the goals of the paper and also individual sections more clear. Some further checks are also needed.

Overall:
I find the paper difficult to read, first of all, because it is overly long and not well organized. I would suggest moving some details to the appendix and adding some sketches to make it easier to understand. It feels like different parts of the paper are written by different people, at least in a very different style with some parts very condensed and others too colloquial. For example, I would not use sentences like: "and we worry that we might have thrown out important parts of the cross correlations". Instead of these kinds of sentences, proper scientific checks should be made to see the level of compromise made when throwing out the majority of edges!

Major:
1. Title: I believe the title is not suitable: it contains three abbreviations that may not be familiar to the readership of the journal. I would suggest grasping the general goals and approach followed rather than enumerating the names of the particular methods. 
2. Lines 354-357: Did you check what happens if the typical gaps in the voids are increased? It should be checked to understand how the algorithm behaves in non-ideal situations.
3. Lines 394-397: For case C4, the cylindrical shell does not enclose a void? Why don't we find \beta_2=1?
4. Lines 410-412: "Ultimately, this model is too complex for us to derive
neat insights from". If the results for D4 cannot be interpreted, how do you interpret the Betti numbers for complex network structures? Can you check by at least a visual inspection that the void indeed disappeared from D3 to D4?
5. Sect. 5.5: It is difficult to unravel which are the exact steps followed here. Include a pseudo-code similar to the one for MST and PMFG algorithms!
6. Include the number of nodes/edges/simplices in Table 1.
7. Line 426: What is this correlation threshold? Don't you use the d_ij distances here?
8. Figure 8: What is on the axes? What is a barcode here?
9. Lines 569-590: Interesting science-historical story but the paper is anyhow too lengthy for including it. Shift it to an appendix.
10. Lines 630-633: If it was introduced for nodes how do you switch to edges? ORC should then be redefined! More explanation is needed. 
11. Figure 10: It is difficult to interpret the results here: colors are not really visible everywhere, maybe the black dots are too large. Which are the ORC values for the leaf of the tree? How for you interpret these values for a tree-like graph compared to the obvious interpretation of Fig. 9?
12. Figs 11,12,13: What is the difference between Fig 11 and Fig 12(b)? What about Fig 12(b) and Fig 13(I)? Only the layouts are changing? It is not very informative. I would remove redundant plots and merge them into a single figure (Figs 11,12,13).
13. For the networks obtained from the geometrical toy models, the ORC values can be used to predict fission/fusion processes. Can you show an example of such events for TWSE networks where it is clearly visible that before/after to process we have one more/one less component and that with time evolution the ORC curvature indeed changes abruptly?
14. Fig 14: What happens for other threshold values than ORC=-0.5? Do we get the same/similar results?
15. References: As the number of references listed here is more suitable for a review paper being still not representative when it comes to important methods of econophysics (like inverse statistics, see the works of Jensen, Simonsen, or Neda) and network theory applied to signal analysis (state-transition networks, recurrence networks, etc), I would either reduce their number to below one hundred or cite other important contributions as well to make it more comprehensive.
16. Discussion: Alternative methods for detecting/predicting phase transitions in markets should be discussed here. See for example how bifurcation points can be predicted by state-transitions networks:
A Novel Measure Inspired by Lyapunov Exponents for the Characterization of Dynamics in State-Transition Networks, Entropy 23 (1), 103, (2021)

Minor:
1. Line 69: PMFG not yet introduced (except for the abstract) but already used as an abbreviation. 
2. Above Eq. (1): How do you/we know about discussions between Sornette and Mantegna? Personal communication/reference?
3. Eq. (1): Does C_ij denote Pearson-type normalized correlations with C_ij \in [-1,1]? Pearson-type cross-correlations are only mentioned on page 21. It should be clarified already at this point.
4. Figure 3: The text Kruskal.png should be removed.
5. Figure 3(a): `if (I,j) not connected` should read: there is no path between nodes (I,j) OR nodes (I,j) are not part of the same tree OR something similar.
6. Figure 3: Maybe a sketch would be useful to understand the procedure better OR provide a little bit of more detailed discussion in the caption/main text.
7. Line 329: The second "the" can be omitted.
8. Line  335: There are no captions/axis labels etc provided with The supplementary materials.
9: Line 335: Barcodes first mentioned here. How to read them? What is their meaning?
10: Figure 5: Too small axis labels.
11. Line 448: closed -> close
12. Line 475: Previously you had d_ij in Eq. (1) and not D_ij.
13. Table 1: Is there a last column? Something missing?
14. Figure 7: Are these results for the seventh time period? Why not mentioned this here but only later on hidden in the main text? D_ij -> d_ij again?
15. Eq. (3): Why are two (v,v) vectors in the argument? 
16. Line 644: What is the \alpha self-transition parameter? Is it important here? 
17: Line 666: Previously you had C_ij instead of C(i,j).

Round 2

Reviewer 2 Report

I appreciate the effort the authors put into the revision of the paper. I believe the structure of the paper is now more logical, many further sketches and examples have been added. 

I still have one minor comment/question related to the computation of the here defined distance D_{ij} in Eq. (2). As far as I see it is not specified at this point what the time series x_i and x_j consist of: are they the prices/daily price differences/daily log-returns of individual stocks (as in Ref. [26]), or some renormalized versions of these? I believe this should not be hidden somewhere in the supplementary materials. Furthermore, checking the code provided with the paper, the line

def process_data(df, window_size, smooth, diff): 
    df = (df - df.median())/df.std()      #Scale Data

seems to renormalize the time series by shifting it with respect to the median value and not the mean. This may be confusing for the readers since most of the literature uses the mean for normalization.

Round 3

Reviewer 2 Report

Thank you for the responses.